# Variations in Victimization: The Relationship between Community Types, Violence against Women and Reporting Behaviors

Ryan Randa [1,*], Sarah R. Bostrom [2], Wyatt Brown [3], Bradford W. Reyns [4] and Jessica C. Fleming [1]

1   College of Criminal Justice, Sam Houston State University, Huntsville, TX 77341, USA; jcc063@shsu.edu
2   Department of Criminology & Justice Studies, Avila University, Kansas City, MO 64145, USA;
    sarah.bostrom@avila.edu
3   Department of Criminology, Sam Houston State University, Huntsville, TX 77341, USA; wbrown@shsu.edu
4   Department of Criminal Justice, Weber State University, Ogden, UT 84408, USA; breyns@weber.edu
*   Correspondence: ryan.randa@shsu.edu

**Abstract:** Existing research suggests that victimization risk is higher among urban residents. Violence against women is a notable exception in this trend. While the literature does indicate that rural women are at equal risk for violent victimization, it does not differentiate between types of non-urban spaces (exurbs, suburbs, small towns, dispersed rural). We use a five-category measure of rural-urban location articulated land use to disentangle victim–offender relationship distribution using a female victim sample from the 1996–2005 United States National Crime Victimization Survey (NCVS). In the most rural areas (dispersed rural locations), women are most likely to be victimized by friends or acquaintances. The proportion of women victimized by strangers in dispersed rural locations is very low. As urbanicity increases, so does the proportion of women victimized by strangers. The findings indicate that victim–offender relationships may be dictated by proximity. In dispersed rural locations, there are comparatively fewer people unknown to the victim than in central city locations. Consequently, proximity dictates that offenders in dispersed rural locations are unlikely to be strangers. The articulated land use measure ensures that the differences between types of rural and suburban locations are identified. Future research should consider the impact of proximity on rural victimization and increased specificity in rural measurements.

**Keywords:** rural; measurement; proximity; violence against women; victim–offender relationship

## 1. Introduction

Violence against women (VAW) continues to persist as a challenge in United States society (Raphael et al. 2019). There are associated issues with measurement and truly understanding the breadth and depth of the problem. One dimension of this difficulty is that violence against women is consistently underreported. Scholars have proposed myriad reasons for underreporting, and research has demonstrated that underreporting is a widespread phenomenon not limited to any one society but is a global issue (Palermo et al. 2014; Viero et al. 2021). The aim of this study is to explore underreporting in the United States through the analysis of data collected in the United States National Crime Victims Survey (NCVS). We will explore underreporting through a lens of urbanicity, age, and relationship to the offender.

We will begin by exploring whether community classification (urban core, suburban, exurb, small town, or dispersed rural) is associated with different victim–offender relationship classifications. This will contribute to the growing body of research on violence against women through the application of conceptual aggregates by community density type recently articulated by DuBois et al. (2019). A large majority of studies that examine crime and victimization in rural spaces employ a simple dichotomy. What is lost in dichotomous

measures of urban and rural, for example, is that small towns across the United States are largely considered rural in this classification, masking the distinct social differences that exist between small towns and dispersed or unincorporated spaces. The expanded measure better captures the differences across a spectrum of community types (see DuBois et al. 2019 for full discussion).

We follow by addressing whether victims of violence against women are differentially reported across community types. Using the United States National Crime Victimization Survey (NCVS) reports on crimes against women—including sexual victimization—we examine the relationship between the victim and the perpetrator by delineation of geographical areas within the United States. We hypothesize that the proportion of offenders known to victims is a function of differential rates of exposure as operationalized by community type. Finally, this study will conclude with an analysis of individual and incident characteristics that may influence the likelihood of reporting a victimization to the police.

For decades, studies have supported the proposition that crime and criminal victimization are a function of exposure to certain victimization risks (e.g., Madero-Hernandez and Fisher 2012; McNeeley 2015; Spano and Freilich 2009). Indeed, the so-called chemistry for crime is a combination of the availability of suitable targets, in conjunction with exposure to motivated offenders, and a decrease or absence in capable guardianship (e.g., Cohen and Felson 1979; Felson and Cohen 1980). In addition, the risky lifestyles perspective on exposure to motivated offenders has been a prominent theory of victimization risk (e.g., Cohen et al. 1981; Hindelang et al. 1978)[1]. Logically, the type of community in which one resides has a direct relationship to risk of victimization via increases in exposure, yet we aim to address whether the associated differences in community type and offender relationship result in discernible differences in reporting VAW to the police.

## 2. Review of Literature

### 2.1. Reporting Victimization to Police

Research on reporting victimization to the police highlights the gap between known victimizations and crime statistics. It is universally accepted by scholars in this field of study that violence against women in all forms, particularly physical and sexual violence, goes unreported and underreported. Particular forms of violence against women, such as physical and sexual violence, show distinct reporting trends (Tjaden and Thoennes 2000; Truman and Morgan 2014). Estimates suggest that as many as one in four women will experience some form of violence in their lifetime, and yet scholars conclude that many of these women will never report the incident to the police (Truman and Morgan 2014). Often referred to as the "iceberg" (see Garcia 2004), reported violence against women, such as domestic violence, can be thought to represent only one small portion of a much greater problem. These unreported cases contribute to a culture of silence that normalizes violence against women and perpetuates the cycle of abuse (Garcia 2004).

Notwithstanding this trend, research has found that the likelihood that a crime is reported to the police is greater for older victims (Baumer 2002; Bosick et al. 2012; Hashima and Finkelhor 1999; Watkins 2005) and women (Baumer and Lauritsen 2010). However, distinct barriers discourage women, especially those in rural areas, from reporting their victimization to the police and receiving support, largely due to long distances, poverty, and gender-role dynamics within couples (Ceccato 2015). These obstacles are often rooted in geographical and social isolation and are further reinforced by community norms that discourage women from seeking support (DeKeseredy 2022; DeKeseredy et al. 2016; Rennison et al. 2013).

In rural communities, lack of privacy and fear of breached confidentiality, coupled with scarce resources such as shelters and support services, can inhibit reporting specifically for cases involving sexual and physical violence against women (DeKeseredy 2019; DeKeseredy et al. 2016; Edwards 2015; Farhall et al. 2020; Logan et al. 2003; Peek-Asa et al. 2011; Perez-Patron et al. 2020; Riffe-Snyder et al. 2021; Swan and Hobbs 2016). Rural women also face amplified fear, stigma, and shame associated with domestic violence, further hindering

reporting (DeKeseredy 2015; Lanier and Maume 2009; Little 2017). In addition, financial and housing dependencies on their abusers, limited employment opportunities, lower incomes, and a lack of affordable housing in rural areas make it difficult for women, especially younger ones, to leave abusive relationships (Rennison et al. 2013). Furthermore, rural women often have fewer social support resources (Krishnan et al. 2001), confront social and geographic isolation (Eastman et al. 2007), have inadequate reliable transportation (Arcury et al. 2005; Logan et al. 2003), and are less likely to be insured, limiting access to healthcare services (Black et al. 2011; Pruitt 2007).

While a number of arguments have been made for social and structural constraints to reporting domestic violence and other forms of violence against women, there is space yet to be explored for violence against adolescent girls—those aged between 12 and 18. A review of the literature suggests that this group may be even more underserved than their adult counterparts (Finkelhor and Ormrod 2001; Finkelhor et al. 2001). Furthermore, studies have revealed a higher risk of victimization among rural adolescent girls than their urban counterparts (Martz et al. 2016; Musu-Gillette et al. 2018; Peek-Asa et al. 2011; Liu et al. 2021).

The most current works In this area of research suggest that the factors associated with reporting to the police are nuanced but can be driven by incident-level factors such as requiring medical attention (Kaylen and Pridemore 2015), having a bystander call the police (Felson et al. 2005), and the relationship to the offender (if any). Structural constraints are most often reported to be applied to VAW when the offender is a current or former intimate partner, family member, or close friend, especially in rural areas (Hamby 2014). Such constraints are often linked to financial and housing dependencies, with victims in rural areas facing additional barriers related to geographical isolation and limited legal support (Logan et al. 2003). Furthermore, social constraints can stem from victims' concerns over changing perceptions within their tight-knit communities, a factor that is particularly prominent in rural areas (DeKeseredy 2022).

Victims often face barriers when reporting incidents to the police, predominantly driven by the fear of negative reactions from their communities, families, and the authorities (Edwards 2015; Hamby 2014). Many of the difficulties associated with reporting to the police are closely connected to the ideology of victim-precipitated violence and the perception that they will not be treated fairly in the criminal justice system and experience unjust repercussions at home and in the community. For example, Hamby (2014) explains that these fears may discourage victims from reporting their experiences, perpetuating cycles of violence and fostering a culture of underreporting. Moreover, this fear is deeply entrenched in the concern that they may be unjustly held accountable for the violence inflicted upon them and may subsequently encounter discriminatory treatment or a lack of adequate support from the criminal justice system.

*2.2. Importance of Community Type*

In discussions of crime, crime trends, and predictors of crime, space matters. Social structures, opportunity, and informal social supports vary with geography and can affect the prevalence and reporting rates of violence against women (Barnett and Mencken 2002; Braithewaite 2015; Deller and Deller 2012; Logan et al. 2003; Rennison et al. 2013; Schwartz and Gertseva 2010). Specifically, the geographical distance and isolation in rural areas exacerbate the problem, making it difficult for neighbors to notice violence (DeKeseredy et al. 2004). Additionally, escape routes are limited due to the distance to women's shelters, lack of public transportation (St. Cyr et al. 2021; Peek-Asa et al. 2011; Websdale 1998), and limited internet or mobile phone access (DeKeseredy and Joseph 2006). Compared to the emphasis on urban crime, less attention has been paid to the unique dynamics of violence against women in other community types, particularly in rural spaces where intimate partner violence and femicide show significant prevalence (Abraham and Ceccato 2022; Armstead et al. 2021; Bachman 1992; Deller and Deller 2012; Spano and Nagy 2005). This may be a function of the predominantly urban crime drop of the 1990s. While rural crime

also declined, it did not decline at the same sharp rate as urban crime, highlighting the need to examine the specific factors influencing violence against women in rural areas (Bachman 1992; Kaylen et al. 2017; Schwartz and Gertseva 2010; Spano and Nagy 2005). Yet, with increasing attention to research in rural communities, there have been reported increases in rural crime, particularly in intimate partner violence—a finding contrary to overall crime trends of the 1990s and 2000s (Barnett and Mencken 2002; Rennison et al. 2013; Schwartz and Gertseva 2010; Spano and Nagy 2005). For example, intimate femicide, the murder of women by current, former, or potential partners, is a crime that is not only increasing but is also notably more prevalent in rural than in urban or suburban regions (DeKeseredy et al. 2016).

According to Bachman (1992), crime outside urban spaces is perceived as non-problematic and not a growing concern. In contrast, Barnett and Mencken (2002) found, in their study of the impact of population change and socioeconomic status on crime rates in metropolitan and nonmetropolitan counties, that rural spaces experienced a slight increase in violent crime during the urban crime drop. It appears, then, that there is differential vulnerability to crime based on community type, which makes the study of crime by community type important, especially when examining violence against women (Bachman 1992).

Collapsing urban, suburban, and rural areas into forced dichotomies may mask variation in trends among these community types. Consequently, a measurement continuum of urban to rural spaces is needed to fully understand the contextual importance of community type in crime rates (see DuBois et al. 2019). In an original study testing this continuum, DuBois et al. (2019) found that there were marked differences in findings regarding intimate partner violence victimization (IPV) and urbanicity based on measurement. Specifically, when an urban/rural dichotomous measure was used, urban spaces demonstrated significantly higher rates of IPV (DuBois et al. 2019). When a three-category (urban, suburban, rural) measure was used, suburban locations had significantly lower rates of IPV compared to the other locations (DuBois et al. 2019). There were no significant differences in IPV between urban and rural spaces (DuBois et al. 2019). Finally, when the expanded articulate land use continuum was used, small towns (designated as urban in a dichotomous measure and rural in a three-category measure) demonstrate significantly higher rates of IPV compared to the other categories (DuBois et al. 2019). Clearly, measurement of urbanicity requires careful consideration, particularly when examining rural spaces. Additionally, in the same spatial location, different data sources may record differing rates of crime (Berg and Lauritsen 2016). There are also special concerns regarding crime reporting and data recording in rural spaces (Berg and Lauritsen 2016). Data reporting categories (jurisdiction or location of victim's residence) also impact crime rates in aggregate locations (Berg and Lauritsen 2016). Therefore, specificity in determining the categorization of the spatial location is especially important.

### 2.3. Exposure as a Function of Community Type

Patterns in victimization across spatial locations may be a function of proximity and exposure to motivated offenders, with violence against women presenting distinct patterns (Tjaden and Thoennes 2000). Cohen et al. (1981, p. 507) define proximity as "the physical distance between areas where potential targets of crime reside and areas where relatively large populations of potential offenders are found". In the aspect of violence against women, this implies that the closer the relationship of the potential victim and perpetrator, the higher the likelihood of an incident occurring (Cohen et al. 1981). In their seminal study, proximity is measured using a seven-category compilation of neighborhood income and community type (central city, non-central city, or small towns and rural areas) (Cohen et al. 1981). They found that residents of rural areas are significantly less likely to be victims of assault, burglary, or personal larceny than residents of other categories (Cohen et al. 1981). However, when looking specifically at violence against women, the pattern shifts; DeKeseredy and Schwartz (2013) found that women in rural areas were more likely to face domestic violence than those in urban areas. Wilcox et al.'s (2003)

multicontextual opportunity theory offers a valuable lens through which to explore these dynamics. It emphasizes that opportunity structures for violence exist at both micro-level (individual and situational) and macro-level (community and societal) contexts. In essence, the structure of the community partially determines the risk for victimization through proximity.

Importantly, proximity and exposure are closely related. Proximity dictates the physical distance between potential victims and motivated offenders, while exposure controls an offender's access to potential victims (Cohen et al. 1981). Across community types, proximity to motivated offenders will vary. Within community types with similar proximity, exposure will determine the victim–offender relationship in a sample of victims. The social structure of the community type, including social ties and social networks, will impact exposure and offender access to potential victims.

Taking a step further, it is critical to acknowledge that the social structure of the community type, including its social ties and social networks, can impact exposure and offender access to potential victims in a significant way. These differences are expected to be largest between the most urban and most rural community types (Cohen et al. 1981; Lee 2000). For example, social ties and social networks are especially strong in rural communities (DeKeseredy 2015; Donnermeyer 2015). Social privacy is low, with residents being involved in each other's lives (Weisheit and Donnermeyer 2000). Dense social networks ensure that most residents in rural communities are known to each other.

Population stability in rural communities ensures easy identification of strangers (Bouffard and Muftić 2006; Donnermeyer 2015). Considering the strong social ties and suspicion of outsiders with residents of rural communities, the protection offered by the anonymity often found in urban spaces is lacking (Weisheit and Donnermeyer 2000). This factor can significantly impact the patterns of violence against women and add complexity to the victim–offender relationship in such environments (Weisheit and Donnermeyer 2000). The close-knit nature of social structures in rural communities often works to the advantage of offenders, facilitating greater access to potential victims and reducing the chances for victims to report or escape from violent situations (Lockwood and Terry 2021; Treat et al. 2022). Proximity and exposure to strangers are less likely in rural communities, meaning offenders may be more likely to be acquaintances rather than strangers when compared to other community types.

### 2.4. The Current Study

The current study will explore the data for disparities in the victim–offender relationship by aggregated community type. After examining sample descriptive statistics (Tables 1 and 2), we conduct a basic assessment of this assumption via bivariate categorical analysis and will present corresponding contingency tables (Tables 3 and 4). In assessing whether there is a significant association, we will employ the extended Cochran–Mantel–Haenszel (Zhang and Boos 1997) tests of bivariate multinomial relationships with stratification. These tests allow, in essence, for control variables to be inserted in the chi-square process (Yu and Gastwirth 2008). In doing so, we will address whether community type corresponds to any disparity in likely offenders. We hypothesize that the proportion of offenders known to victims is related to rates of exposure based on community type. After addressing the question of disparity in victim–offender relationships by community type, we will explore the relationship between community type and non-reporting of acts of VAW. Finally, we will explore the relative probabilities of non-reporting in a multivariate context that accounts for community, individual, and incident characteristics (Tables 5 and 6).

**Table 1.** Land use measures and sample distribution.

| | | | | DuBois et al. (2019) | |
|---|---|---|---|---|---|
| Urban | 81.7% | City of MSA | 38.3% | Urban Core | 38.3% |
| | | | | Suburb | 37.1% |
| | | (S)MSA no | 47.5% | Exurb | 6.3% |
| Rural | 18.3% | | | Small Town | 10.4% |
| | | Not (S)MSA | 14.2% | Dispersed Rural | 7.8% |
| | 100.0% | | 100.0% | | 100.0% |

**Table 2.** Descriptive statistics, estimation sample (Number of obs: 16,810).

| Variable | Proportion | Std. Dev. | Min | Max |
|---|---|---|---|---|
| Non-Report | 0.506 | 0.500 | 0 | 1 |
| **Community Type** | | | | |
| Dispersed Rural | 0.078 | 0.269 | 0 | 1 |
| Small Town | 0.104 | 0.305 | 0 | 1 |
| Exurbs | 0.064 | 0.244 | 0 | 1 |
| Urban Core | 0.383 | 0.486 | 0 | 1 |
| **Victim–Offender Relationship** | | | | |
| Known Offender | 0.020 | 0.139 | 0 | 1 |
| Family Member | 0.005 | 0.068 | 0 | 1 |
| Stranger | 0.005 | 0.068 | 0 | 1 |
| Intimate Partner | 0.020 | 0.142 | 0 | 1 |
| Multiple Offenders | 0.055 | 0.227 | 0 | 1 |
| Known from Work | 0.002 | 0.041 | 0 | 1 |
| **Age ~ reference category 27–35** | | | | |
| 12–18 years | 0.232 | 0.422 | 0 | 1 |
| 19–26 years | 0.222 | 0.416 | 0 | 1 |
| 45+ years | 0.334 | 0.472 | 0 | 1 |
| **Victim Educational Attainment (High school or less = 0)** | | | | |
| Attended College | 0.375 | 0.484 | 0 | 1 |
| Bachelor's Degree+ | 0.030 | 0.170 | 0 | 1 |
| Sexual Victimization | 0.086 | 0.281 | 0 | 1 |
| Weapon Present | 0.200 | 0.400 | 0 | 1 |
| Medical Care Necessary | 0.122 | 0.327 | 0 | 1 |
| White Victim | 0.820 | 0.385 | 0 | 1 |
| Non-White Offender | 0.463 | 0.499 | 0 | 1 |

**Table 3.** Distribution of perpetrator type by land use: observed and column percent.

| | Land Use (DuBois et al. 2019) | | | | | |
|---|---|---|---|---|---|---|
| | Dispersed Rural | Small Town | Exurb | Suburb | Urban Core | Total |
| **Perpetrator** | | | | | | |
| Intimate Partner | 302 | 364 | 284 | 1314 | 1198 | 3462 |
| | 22.72 | 20.62 | 26.42 | 20.92 | 18.47 | 20.44 |
| Family Member | 110 | 159 | 84 | 401 | 402 | 1156 |
| | 8.28 | 9.01 | 7.81 | 6.39 | 6.2 | 6.83 |
| Acquaintance | 610 | 742 | 464 | 2,375 | 2,417 | 6608 |
| | 45.9 | 42.04 | 43.16 | 37.82 | 37.26 | 39.02 |
| Stranger | 307 | 500 | 243 | 2190 | 2470 | 5710 |
| | 23.1 | 28.33 | 22.6 | 34.87 | 38.08 | 33.72 |
| **Total** | 1329 | 1765 | 1075 | 6280 | 6487 | 16,936 |

Pearson chi2(12) = 229.2841 Pr = 0.000
likelihood-ratio chi2(12) = 235.7932 Pr = 0.000

**Table 4.** Distribution by articulated land use: observed, expected, and chi-square contribution.

| | Land Use (DuBois et al. 2019) | | | | | |
| | Dispersed Rural | Small Town | Exurb | Suburb | Urban Core | Total |
|---|---|---|---|---|---|---|
| **Perpetrator** | | | | | | |
| Intimate Partner | 302 | 364 | 284 | 1314 | 1198 | 3462 |
| | 271.7 | 360.8 | 219.7 | 1283.7 | 1326.1 | 3462.0 |
| | 3.4 | 0.0 | 18.8 | 0.7 | 12.4 | 35.3 |
| Family Member | 110 | 159 | 84 | 401 | 402 | 1156 |
| | 90.7 | 120.5 | 73.4 | 428.7 | 442.8 | 1156.0 |
| | 4.1 | 12.3 | 1.5 | 1.8 | 3.8 | 23.5 |
| Acquaintance | 610 | 742 | 464 | 2375 | 2417 | 6608 |
| | 518.5 | 688.7 | 419.4 | 2450.3 | 2531.1 | 6608.0 |
| | 16.1 | 4.1 | 4.7 | 2.3 | 5.1 | 32.5 |
| Stranger | 307 | 500 | 243 | 2190 | 2470 | 5710 |
| | 448.1 | 595.1 | 362.4 | 2117.3 | 2187.1 | 5710.0 |
| | 44.4 | 15.2 | 39.4 | 2.5 | 36.6 | 138.1 |
| **Total** | 1329 | 1765 | 1075 | 6280 | 6487 | 16,936 |
| | 1329.0 | 1765.0 | 1075.0 | 6280.0 | 6487.0 | 16,936.0 |
| | 68.0 | 31.7 | 64.4 | 7.3 | 57.9 | 229.3 |

**Table 5.** Logistic regression on non-reporting.

| | Coef. | Std. Err. | z | P > z | AOR |
|---|---|---|---|---|---|
| **Community Type** | | | | | |
| Dispersed Rural | −0.144 | 0.064 | −2.25 | 0.0250 | 0.8656 |
| Small Town | −0.096 | 0.057 | −1.68 | 0.0930 | 0.9081 |
| Exurbs | −0.180 | 0.070 | −2.57 | 0.0100 | 0.8353 |
| Urban Core | −0.073 | 0.038 | −1.92 | 0.0540 | 0.9295 |
| **Victim–Offender Relationship** | | | | | |
| Known Offender | 0.509 | 0.121 | 4.19 | 0.0000 | 1.6630 |
| Family Member | 0.440 | 0.251 | 1.75 | 0.0790 | 1.5534 |
| Stranger | 0.029 | 0.235 | 0.12 | 0.9020 | 1.0294 |
| Intimate Partner | 0.130 | 0.116 | 1.12 | 0.2630 | 1.1383 |
| Multiple Offenders | −0.253 | 0.076 | −3.33 | 0.0010 | 0.7761 |
| Known from Work | 1.345 | 0.451 | 2.98 | 0.0030 | 3.8373 |
| **Age ~ reference category 27–35** | | | | | |
| 12–18 years | 1.130 | 0.053 | 21.27 | 0.0000 | 3.0965 |
| 19–26 years | 0.275 | 0.049 | 5.58 | 0.0000 | 1.3164 |
| 45+ years | 0.132 | 0.045 | 2.92 | 0.0030 | 1.1411 |
| **Victim Educational Attainment (High school or less = 0)** | | | | | |
| Attended College | 0.376 | 0.037 | 10.12 | 0.0000 | 1.4566 |
| Bachelor's Degree+ | 0.653 | 0.101 | 6.44 | 0.0000 | 1.9214 |
| Sexual Victimization | 0.681 | 0.061 | 11.1 | 0.0000 | 1.9758 |
| Weapon Present | −0.652 | 0.042 | −15.52 | 0.0000 | 0.5209 |
| Medical Care Necessary | −1.026 | 0.054 | −19.08 | 0.0000 | 0.3584 |
| White Victim | 0.182 | 0.047 | 3.88 | 0.0000 | 1.1994 |
| Non-White Offender | −0.100 | 0.037 | −2.67 | 0.0080 | 0.9052 |
| Year | −0.047 | 0.005 | −10.22 | 0.0000 | 0.9539 |
| Constant | 93.917 | 9.222 | 10.18 | 0.0000 | |
| Logistic regression | Number of obs | = | 16,810 | | |
| | LR chi2(21) | = | 1700.33 | | |
| | Prob > chi2 | = | 0.000 | | |
| Log likelihood = −10,800.519 | Pseudo R2 | = | 0.073 | | |

**Table 6.** Predicted probabilities of non-reporting.

|  | Probability | Std. Err. | z | P > z |
|---|---|---|---|---|
| Community Type |  |  |  |  |
| Dispersed Rural | 0.484 | 0.015 | 33.13 | 0.000 |
| Small Town | 0.496 | 0.013 | 38.90 | 0.000 |
| Exurbs | 0.475 | 0.016 | 29.34 | 0.000 |
| Suburb | 0.520 | 0.007 | 77.83 | 0.000 |
| Urban Core | 0.502 | 0.007 | 74.87 | 0.000 |
| Victim–Offender Relationship |  |  |  |  |
| Unknown | 0.504 | 0.004 | 116.91 | 0.000 |
| Known Offender | 0.628 | 0.028 | 22.37 | 0.000 |
| Family Member | 0.612 | 0.059 | 10.29 | 0.000 |
| Stranger | 0.511 | 0.059 | 8.72 | 0.000 |
| Intimate Partner | 0.536 | 0.028 | 18.83 | 0.000 |
| Multiple Offenders | 0.441 | 0.018 | 24.27 | 0.000 |
| Known from Work | 0.796 | 0.073 | 10.86 | 0.000 |
| Age Category |  |  |  |  |
| 12–18 years | 0.686 | 0.008 | 84.18 | 0.000 |
| 19–26 years | 0.481 | 0.009 | 56.04 | 0.000 |
| 27–44 years | 0.414 | 0.009 | 47.90 | 0.000 |
| 45+ years | 0.446 | 0.007 | 62.85 | 0.000 |
| Victim Educational Attainment |  |  |  |  |
| High School Diploma or less | 0.465 | 0.006 | 84.31 | 0.000 |
| Attended College | 0.558 | 0.007 | 81.13 | 0.000 |
| Bachelor's degree+ | 0.625 | 0.023 | 27.23 | 0.000 |
| Sexual Victimization |  |  |  |  |
| No | 0.490 | 0.004 | 115.52 | 0.000 |
| Yes | 0.655 | 0.013 | 49.24 | 0.000 |
| Weapon Present |  |  |  |  |
| No | 0.537 | 0.005 | 119.31 | 0.000 |
| Yes | 0.377 | 0.009 | 42.50 | 0.000 |
| Medical Care Necessary |  |  |  |  |
| No | 0.536 | 0.004 | 125.22 | 0.000 |
| Yes | 0.293 | 0.011 | 27.82 | 0.000 |
| White Victim |  |  |  |  |
| No | 0.467 | 0.010 | 44.91 | 0.000 |
| Yes | 0.513 | 0.005 | 112.16 | 0.000 |
| Non-White Offender |  |  |  |  |
| No | 0.516 | 0.006 | 86.91 | 0.000 |
| Yes | 0.491 | 0.006 | 76.21 | 0.000 |

## 3. Data and Methods

The data being explored in the present study are available publicly and can be identified as the combined National Crime Victimization Survey (NCVS) 1992–2005. We chose to use the NCVS data for this project in part because it has been used for the purpose of exploring victimization across community types in the past, partly because of the aggregated form, which combines multiple years of collection efforts, but also because it addresses crimes and victimizations that may not have been reported to the police. This arguably makes it a superior choice among the publicly available official data collections for addressing domestic violence and other underreported crimes. It includes female respondents who experienced violent victimization, including threatened, attempted, and completed rape, sexual assault, robbery, aggravated assault, and simple assault.

*Measures*

Our primary variable is a dichotomous measure addressing non-reporting to the police. The NCVS directly asks respondents whether or not they contacted the police in response to a victimization, and we focus on this element. We code this such that not contacting the police is the positive outcome (1 = did not contact the police). Other measures in the NCVS address additional indirect mechanisms for eventual law enforcement contact, such as whether another person contacted the police. However, our focus is strictly on the victim's direct intentional contacting of the police.

An additional key element of interest here is the nature of the relationship between the victim and the offender. The NCVS captures this information in as many as 18 different categorizations. For our purposes, these relationships are grouped into categories, including (1) current and former intimate partners (spouse, ex-spouse, current and former boyfriend/girlfriend); (2) family members (parent or step-parent, child or step-child, sibling, or other relative); (3) known persons that are not family or extended family (friends, neighbors, schoolmates, etc.); (4) strangers or persons unknown to the victim; (5) multiple offenders; and (6) work contacts (clients, customers, and co-workers). In this sample, when considering only the cases where a relationship is described, the majority of perpetrators were classified as acquaintances. Thirty-nine percent (39%) of victims were acquainted with the offender in some way, while another 34% did not know their attacker, and 20% of respondents identified their attacker as a current or former intimate partner. For analysis, this measure is categorical.

Secondly, and equally important, is the measurement of settlement type. The current study will employ a recently devised categorical measure of land use, or settlement areas, which further delineates rural and urban communities. DuBois et al. (2019) assert that the commonly used MSA measure attributed to the Office of Management and Budget in the NCVS is inappropriate for identifying urban, suburban, and rural locations. Instead, they propose a new six-category measure that more specifically captures variation along the urban-to-rural location continuum (DuBois et al. 2019). This new measure of community type uses a combination of the two available measures of land use found in the NCVS (see DuBois et al. 2019). We apply this measure to the data, as well as their source components—the traditional MSA and rural/urban dichotomy—in an attempt to evaluate the utility of employing further articulated measures of community type.

The measure suggested by DuBois et al. (2019) creates a 6-category community type. Settlements designated as urban by both the land use and MSA measures are defined as the "urban core". Those considered urban in the land use measure and MSA but not city in the MSA measure are "suburban areas". Areas that are urban in the land use measure and non-MSA are labeled "small towns". Additionally, settlements that are rural in the land use measure and considered a central city of MSA in the MSA measure are "enclaves" and most rare. Rural areas in the land use variable that are not cities but are in MSA are "exurban". Finally, settlements identified as rural in the land use measure and non-MSA in the MSA measure are designated "dispersed rural" in the new measure. Enclaves, category 5, were dropped due to too few cases (see Table 1).

Additional information describing the sample includes age, race of the offender, race of the victim, educational attainment, whether the victimization was sexual in nature, whether a weapon was present, and whether medical attention was necessary (see Table 2). Age is a categorical variable featuring adolescents (12–18), young adults (19–26), and adults (27–44) as principal interests. Educational attainment is categorical and condenses a wide range of possible responses; we have condensed the measure to a three-category measure: high school diploma (or equivalent) or less, college experience—including having earned an associate's degree, and a bachelor's degree or higher. The race of the victim is measured dichotomously (white only). The race of the offender is measured dichotomously as Non-white. The remaining measures are dichotomous, where the associated name is coded positively (name = 1). Finally, for reference, we have included in an appendix (Appendix A

Table A1) the list of crimes IIded in the violence against women measure and the associated frequencies.

## 4. Results

Results of the analysis, beginning in Tables 3 and 4, present an identifiable disparity in the distribution of victim–offender relationships across community designations. Table 3 contains an unweighted contingency matrix between the victim–offender relationship and community designations. This matrix also includes the column percentages, such that victim–offender relationship is distributed by community. Chi-square and log-likelihood tests indicate a significant association. Table 4 further presents information regarding the relationship between community type and the victim–offender relationship by including information on the gaps between the expected and observed frequencies, as well as the chi-square value added for each matrix cell. These gaps in observed and expected frequencies enable us to begin to determine whether certain contingencies are occurring (or not) in a disparate manner.

The largest identifiable gaps occur when the offender is a stranger. The 'urban core' has more reported crimes by strangers than expected. Additionally, both the 'exurban area' and 'dispersed rural spaces' have fewer reported crimes by strangers than would be expected. This expanded measure provides clarity in comparison to Table A2 in the Appendix A, which compares the distribution of perpetrator type using a three-category (urban, suburban, and rural) measure. In Table A2, the areas considered suburban (sMSA not city) demonstrate minuscule differences in victim–offender relationships from what was expected. In contrast, in Tables 3 and 4, exurbs (part of the sMSA location in Table A2) have significantly fewer stranger victimizations than expected. The expanded measure offers increased precision in the distribution of victim–offender relationships by community designation. In reconstructing the table and examining the weighted data, the patterns that emerge are consistent, and the association remained significant in the Cochran–Mantel–Haenszel tests, including the potential confounders listed in Table 2. The chi-square and log-likelihood measures are significant in each of the contingency tables, and both reflect the concentration of stranger-perpetrated crime in urban and suburban spaces. The results presented here confirm the first assumption that a clear picture of victim–offender relationships would emerge with a better articulation of community differences.

As we move from rural vs. urban, one noteworthy difference in the proportional chi-square contribution suggests that when measured as rural vs. urban, the contingency of stranger-perpetrated offenses in rural spaces contributed over half of the total chi-square value (51%). This contribution is a result of the gap in expected and observed frequencies where stranger perpetration in rural spaces was observed far less than expected. When disaggregated (see Tables 3 and 4), the total contribution is 19% of the chi-square value. There are two notable conclusions. First, that stranger-perpetrated violence against women is the source of the largest gaps in expected vs. observed frequencies, regardless of which community-type measure is used. Second, the distribution of perpetrator type is not uniform across community type. Table 4 demonstrates a greater-than-expected frequency of perpetrations by intimate partners in exurban areas, by family members in small towns, and by acquaintances in dispersed rural spaces.

When moving to explore non-reporting, we find that there are differences by community type that persist even after controlling for potential confounders. As noted previously, the final element of this study is a multivariate logistic regression. Diagnostic procedures establish that our model is properly fit and does not require further specification (basic "link test" statistics indicate good model fit and good specification _hat P > |z| = 0.000; _hatsq P > |z| = 0.571). Furthermore, the Hosmer and Lemeshow goodness of fit statistic (collapsed quantiles of probabilities) resulted in a chi-square of 10.73. In testing for potential collinearity issues within our models, we found the mean variance inflation factor (VIF) was 1.11, and no variables posed a VIF above 1.3. Finally, a visual examination for potential

outliers (plotted predicted probabilities against the standardized residuals) revealed no problematic cases.

Table 5 presents the logistic regression analysis results, and we believe that the predicted probabilities in Table 6 provide even greater context and interpretability of the results. For example, Table 6 provides the predicted probability of non-reporting, and we find that at the mean, the predicted non-report probability of a VAW victim residing in the suburbs is 0.52. And, while the probability of not contacting the police is relatively closely clustered across community types, the suburban group produced the highest value. Those in the exurbs and dispersed rural spaces were the least likely to not report their victimization experience to the police. Thus, when we see that dispersed rural and exurban variables produce statistically significant coefficients, they are relative to the reference group (suburban).

With respect to the victim–offender relationship, we find that in this sample, work relationships followed by other known offenders and family generate the highest probability of non-reporting (Table 6). Furthermore, after controlling for potential confounders, victims of VAW at work are roughly 280% more likely to not report relative to the reference group, and those victimized by multiple offenders are significantly less likely to not contact the police. Both findings regarding our key areas of exploration, community type, and victim relationship to offender confirm that they are entwined as it relates to contacting the police, even after accounting for a number of factors meant to address the seriousness of the crime and structural barriers.

Finally, we find that a number of individual and situational factors are potent covariates. When medical care was necessary, a weapon was present, or the offender was Non-white, we find a significant negative relationship, indicating that in these circumstances, a victim is more likely to contact the police. White victims are 20% more likely to not contact the police, and when the victimization is sexual in nature, the victims are significantly less likely to contact the police. Moreover, we find that relative to adults (27–44), all other groups are significantly less likely to contact the police. Most notably, adolescent victims (age 12–18) are 290% more likely to not report their victimization to the police.

## 5. Discussion

The present study examined a central issue in the study of violence against women. Specifically, the influences on the likelihood of personally reporting a victimization across community types and the victim–offender relationship. We investigated how the victim–offender relationship varies across communities in violent crimes against women and hypothesize that community type dictates opportunity structures for such crimes. That is, particular community types should offer greater or lesser opportunities for victimization across victim–offender relationships, with urban settings exposing opportunities for stranger victimization, whereas rural and suburban settings will facilitate perpetration by those with closer relational distance (e.g., family member, intimate partner). Furthermore, we explore how these factors may relate to victims directly reaching out to the police. The results of the analyses highlight several noteworthy observations pertaining to these issues.

In alignment with prior research (Finkelhor and Ormrod 2001; Finkelhor et al. 2009; Truman and Langton 2014), our findings underscore the distressing trend of a significantly higher likelihood of non-reporting among adolescent victims aged 12–18. In fact, adolescents within our sample were an alarming 290% more likely to not report their victimization to authorities compared to their adult counterparts aged 27–44. This glaring disparity suggests the presence of unique, age-related barriers that may deter young victims from reporting their experiences of victimization. Indeed, research has found that adolescent girls may struggle to comprehend the severity of the crime committed against them or have difficulty articulating their victimization experiences (Finkelhor et al. 2015). They also frequently grapple with feelings of shame, embarrassment, fear of retaliation, or fear of not being believed, which can deter them from reporting (Hamby et al. 2013).

A noteworthy observation from our anIlysis indicates that the likelihood of calling the police is not evenly or randomly distributed across community types. Variations in social structures impact proximity, exposure, and guardianship, allowing for different opportunity structures and perhaps different barriers to help-seeking. Wilcox et al.'s (2003) multicontextual opportunity theory examines opportunity from both micro-level and macro-level perspectives. Exposure to motivated offenders at the micro level is determined by lifestyle activities and is related to proximity and accessibility to offenders (Wilcox et al. 2003). On the macro level, exposure varies based on population density and residential and travel patterns of groups of offenders (Wilcox et al. 2003). Specifically, areas with high concentrations of resident offenders experience increased exposure. Places through which or to which large concentrations of offenders travel also have higher exposure. In this perspective, rural spaces have lower exposure on aggregate when compared to other community types (especially urban spaces).

Further, our results highlight that, in rural areas and other community types with less aggregate exposure to motivated offenders, accessibility to some types of offenders will be greater. In dispersed rural spaces, we found that women were significantly less likely to be victimized by strangers than in other community types, yet more likely than their suburban counterparts to call the police. To a degree, this fits with research on rural contexts whereby strangers do not have access to increasingly rural spaces due to high levels of social ties and distrust of strangers (DeKeseredy 2015; Donnermeyer 2015; Weisheit and Donnermeyer 2000). Known offenders do have proximity, exposure, and accessibility to female targets in dispersed rural spaces. This is reflected in the higher-than-predicted proportion of women victimized by acquaintances. Yet, when the offender is known, the victim is less likely to report to the police. The conflation of these findings suggests a need for further exploring the intersectionalities of VAW and contacting the police.

Another point to note is that in the urban core, residents are in proximity and exposed to strangers at higher rates than other community types. The finding regarding increased victimization by strangers in the urban core is congruent with this assessment. While the differences in victim–offender relationships vary most between the urban core and dispersed rural categories, the other community types demonstrate enough variation to support the use of the expanded community-type measure. This expanded measure of community type further articulates violence against women by community type. Specifically, the articulated land use measure unpacks the types of communities that contribute to the percent in chi-square contributions for the more commonly used measures (land use and (S)MSA). For example, while the land use rural category contributes a total of 81.7% of the chi-square value in the land use contingency table, the articulated land use contingency table indicates that this contribution is driven primarily by the dispersed rural (29.7%) and exurban (28.1%) categories in the articulated land use table. Current measures of land use in the study of social issues often rely on limited designations of urban and rural that may not be precise enough to capture the intended population. Whether it is important to address issues within rural communities, explore suburban issues, or examine the nature of crime and victimization in sparsely populated areas, the precision of the measure is a critical element. In many respects, community dynamics are a function of both size and location, where size refers to total population and population density, and location refers to proximity to other communities. As such, exactly how the nature of a place and the characteristics of its location will influence its residents are important parts of social science research.

### 5.1. Limitations

Despite the many strengths of this study, there are limitations that require acknowledgment. Indeed, this study is limited in its ability to demonstrate that the differences in offender type that exist across the aggregates would remain statistically disproportionate after multivariate analysis. Additional analysis at different levels of aggregation is needed to determine the robustness of these findings. There are also limitations to the data itself

as it pertains to the measures of victimization (see Rennison et al. 2013 for discussion). Using the NCVS better captures violence against women than other data sources. However, violence against women may still be underreported in the data. Lastly, the NCVS measurement of community type describes the location of residence of the victim rather than the location of the victimization. We assume that the location of the victimization incidents was correctly captured by the instrument or, at the very least, that the associated error is randomly distributed.

*5.2. Implications and Future Research*

Future research endeavors should continue to explore disaggregation into community type, placing special attention on rural and small-town designations, which traditionally have not been the focal point of many studies. In this regard, it becomes clear that diverse community landscapes—urban, suburban, and rural—each manifest distinct challenges that influence the reporting of victimization (DeKeseredy 2022; DeKeseredy et al. 2016; Rennison et al. 2013). As victims seeking support services may navigate an array of obstacles, such as transportation issues, childcare commitments, and language discrepancies (Overstreet and Quinn 2013), it is imperative that our understanding of these unique contexts and their respective challenges is nuanced and comprehensive. In a bid to effectively navigate these multifaceted challenges, policymakers and practitioners could consider devising support services that are culturally responsive, geographically accessible, and attuned to the specific needs of individual communities. Further, it is essential to implement culturally sensitive public awareness campaigns that utilize age-appropriate messaging to address psychosocial deterrents, including shame, fear of retaliation, and a deficit of knowledge pertaining to available resources.

In addition, future applications of the opportunity perspective on crime may do well to consider aggregation by commonly designated place types. The opportunity perspective and related views on crime and crime prevention have produced innumerable case studies of successful crime prevention interventions, yet there are limitations to the case study. The type of aggregation found here could lead to more generalizable insights and place—meso applications of the theory.

This study highlights the importance of measurement and thoughtful disaggregation of community types. Applications focusing on rural communities and crime should consider the measurement of rural spaces. More specific measurements will allow for greater disaggregation and a better understanding of the contextualizing forces of rural community types. Small towns and dispersed rural spaces are unique contexts with differing structures that should be recognized. Future rural criminology research should re-examine rural crime at additional levels of disaggregation to lead to new insights into the rural experience.

**Author Contributions:** The conceptualization for the project was provided by R.R., S.R.B. and W.B. The methodology is credited to R.R. and S.R.B. R.R. curated the data and completed the formal analysis. The writing—original draft preparation was completed by R.R., S.R.B., B.W.R. and J.C.F. The writing—review and editing, was completed by S.R.B., J.C.F. and R.R. All authors have read and agreed to the published version of the manuscript.

**Funding:** This research received no external funding.

**Institutional Review Board Statement:** Not applicable.

**Informed Consent Statement:** Not applicable.

**Data Availability Statement:** Data are available in a publicly accessible repository. The data presented in this study are openly available online through the Inter-university Consortium for Political and Social Science Research (ICPSR) at https://doi.org/10.3886/ICPSR37241.v2. Accessed on 30 August 2020.

**Conflicts of Interest:** The authors declare no conflict of interest.

## Appendix A

**Table A1.** Violence against women.

| Crime | Frequency |
|---|---|
| Completed rape | 530 |
| Attempted rape | 309 |
| Sexual assault with serious assault | 72 |
| Sexual assault with minor assault | 59 |
| Robbery with injury serious assault | 141 |
| Robbery with injury minor assault | 262 |
| Robbery without injury | 670 |
| Attempted robbery with injury serious assault | 34 |
| Attempted robbery with injury minor assault | 90 |
| Attempted robbery with assault | 346 |
| Aggravated assault with injury | 949 |
| Attempted aggravated assault with weapon | 837 |
| Threatened assault with weapon | 1130 |
| Simple assault with injury | 2799 |
| Sexual assault without injury | 221 |
| Unwanted sex without force | 111 |
| Assault without weapon, without injury | 3750 |
| Verbal threat rape | 91 |
| Verbal threat sex assault | 78 |
| Verbal threat assault | 4545 |
| Total | 17,024 |

**Table A2.** Distribution by Standard Metropolitan Statistical Area: observed, expected, and chi-square contribution.

| | Not sMSA | sMSA Not City | City of sMSA | Total |
|---|---|---|---|---|
| Perpetrator | | | | |
| Intimate Partner | 586 | 1678 | 1198 | 3462 |
| | 491.4 | 1644.5 | 1326.1 | 3462.0 |
| | 18.1 | 0.7 | 12.4 | 31.3 |
| Family Member | 194 | 560 | 402 | 1156 |
| | 164.1 | 549.1 | 442.8 | 1156.0 |
| | 5.5 | 0.0 | 3.8 | 9.4 |
| Acquaintance | 1074 | 3117 | 2417 | 6608 |
| | 938 | 3139.0 | 2531.1 | 6608.0 |
| | 19.7 | 0.0 | 5.1 | 25.0 |
| Stranger | 550 | 2690 | 2470 | 5710 |
| | 810.5 | 2712.4 | 2187.1 | 5710.0 |
| | 83.7 | 0.0 | 36.6 | 120.5 |
| **Total** | 2404 | 8045 | 6487 | 16,936 |
| | 2404.0 | 8045.0 | 6487.0 | 16,936.0 |
| | 127.1 | 1.2 | 57.9 | 186.2 |

Pearson chi2(12) = 186.2 Pr = 0.000
likelihood-ratio chi2(6) = 193.8409 Pr = 0.000

## Notes

[1]   There remains a degree of bifurcation here, as the routine activity approach continues to attract attention as a macro-level explanation of crime trends (e.g., Ashby and Tompson 2017; Brady et al. 2016; Holt et al. 2018; Song et al. 2016; Spano and Freilich 2009), while also being used to identify risk factors for victimization at the individual level.

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
