# Peer review of "Variations in Victimization: The Relationship between Community Types, Violence against Women and Reporting Behaviors"

_socsci, doi:10.3390/socsci12090471_

Round 1
Reviewer 1 Report
This manuscript explores an interesting and important avenue of research by testing the distributive frequencies of violence against women across community type. The paper utilizes a rich archival dataset and explores a new measure for community type. I hope you find my comments helpful.
The methodological focus of the paper is on violence against women, but the literature review focuses on offender-victim trends in general (except for a brief introduction in first couple of paragraphs). Research on violence against women (and issues around non-reporting) stems from distinct theoretical frameworks and the distinction between sexual and physical violence against women is also derived from varied theoretical perspectives. This raises several questions: first, why focus on violence against women for this study as opposed to all instances of reported and non-reported of violence across community-type? Why include both physical and sexual violence against women? Were the authors expecting specific distributions of actual and expected reporting violence against women by community type? If so, why and according to which theories? Hypotheses are not presented until the discussion, when the authors state, “… hypothesize that community type dictates opportunity structures for such crimes. That is, particular community types should offer greater or lesser opportunities for victimization across victim-offender relationships, with urban settings exposing opportunities for stranger victimization, whereas rural and suburban settings will facilitate perpetration by those with closer relational distance (e.g., family member, intimate partner).” However, these directional predictions are not built into the method and results, and in general, it is unclear if this research is testing hypotheses or is only investigating exploratory research questions.
Relatedly, to justify the importance and utility of using the new categorizations of community-type, it would be helpful to set up the importance of the new categorizations in the literature review and why we should expect differences in frequencies and reporting among the new categories as opposed to grouping them all into a rural v. urban category. The literature review focuses on explanations for differences in frequencies of crime and reporting among urban v. rural settings, but what is novel and pertinent for this study is setting up why we need more subcategories for these community-types.
To further evaluate the utility of the new land use measure, it would be helpful to compare the pattern of results for dichotomous measure versus the new measure beyond presenting the difference in descriptives in Table 1. By only presenting the results for the new measure, it is unclear what nuance or extra information this measure is providing above and beyond using the dichotomy for analyses.
Overall, the manuscript would benefit from a more focused theoretical framework that that ties the emphasis on violence against women to the importance and utility of using a new measure of land use.
Author Response
Thank you very much for your time and effort in making comments that improve the manuscript. We've tried to organize our responses and highlight our revisions that align with the reviewers suggestions below.
1. Lit review should be more focused (encompasses VAW, victim-offender trends/ distribution, and non-reporting)
a. Revised the literature review to better align with the study's focus on violence against women. Throughout the literature review, additional sources and information that specifically discuss various aspects of violence against women, ensuring a comprehensive examination of both physical and sexual violence,. were added. Additionally, in the revisions, an effort was made to continuously tie back the information to violence against women, ensuring the literature review consistently emphasizes the main theme of the manuscript.
2. Hypothesis should be highlighted in current study section
a. Added hypothesis to current study section on p. 5
3. Justify importance of expanded community-type measure
a. Added paragraph to discuss findings in DuBois et al. 2019 on p. 4. Essentially, the articulated land use measure demonstrated significantly higher rates of IPV in small towns (part of the rural designation). This finding was obscured using a three-category measure (urban, suburban, and rural) and lacked specificity using a dichotomous measure (rural consolidates small towns and dispersed rural locations).
b. Directed reader to “see DuBois et al., 2019” for a full discussion of the measurement issue on p. 1.
4. Present comparisons (beyond descriptive statistics) of articulated land use measure and others
a. Added table to appendix (Table A2) that compares distribution of victim-offender relationships using the three-category measure of community type on p. 14.
b. Added comparison of Tables 3 & 4 to Table A2 in results section on pp. 7-8. Essentially, the articulated land use measure provides additional clarity particularly regarding exurbs, a location with divergent categorization in the other measures.
Thank you again for your comments
DeKeseredy, W. S., Dragiewicz, M., Rennison, C. M. (2013). Racial/ethnic Variations In Violence Against Women: Urban, Suburban, and Rural Differences. International Journal of Rural Criminology, 2(1), 184-202. https://doi.org/10.18061/1811/53699
Reviewer 2 Report
I have carefully reviewed this manuscript and below is my decision.
- The topic is quite interesting, however, the explanation on the originality of the study is insufficient. This paper, needs to highlight clearly the originality of the study.
- How does the paper contribute to the extant literature on the subject?
- I consider that the paper's methodology is built on the basis of an appropriate theory of logistic regression. However, additional work could be done to improve the methodological arguments as follows:
- Despite the usefulness of the paper, it should undertake several diagnostic tests to examine basic logistic regression assumptions. In logistic regression, basic assumptions must be met, such as the independence of errors, the absence of multicollinearity and the lack of outliers. This can be easily achieved using any statistical or econometric software to enhance the quality of the paper. At present, the paper assumes all these assumptions are fulfilled. Although the findings can still be used where assumptions were violated, this could be explained to the readership.
- The other similar limitation of the paper is the assumption of linearity between the dependent variable and independent variables. This should be tested empirically and completed as a minor revision using diagnostic statistics to prove the linear relationship. Moreover, when the relationship is not linear, the regression findings should also be amended.
- The scale of the measurement should also be discussed more rigorously and clearly, such as which variables are on continuous scales and which ones are dichotomous, nominal, ordinal, interval or ratio. At the moment, the information for the readership on the data is limited.
-I would suggest adding to the literature and referencing it within the introduction and discussion as well. There are studies that have examined intimate partner violence.
1-) https://doi.org/10.1186/s12905-021-01333-1 2-) https://doi.org/10.1371/journal.pone.0275950
It can be published after corrections are made.
Minor editing of English language required
Author Response
I'd like to thank the reviewer very much for highlighting the absence of a discussion on the logistic regression diagnostics in this manuscript. In response to the commentary we have included in the manuscript a limited, albeit pointed, presentation of our efforts in three main areas. First model fit, in order to conduct our analysis we use Stata software packages, and as the reviewer points out the diagnostics and goodness of fit tests are easily accessible in many software packages, including Stata. In the course of establishing our model fit we ran all the fit statistics provided in the “fitstat” command as well as the Hosmer and Lemeshow goodness of fit statistic (lfit) ~ collapsed quantiles of probabilities resulting in a chi square of point 10.73. Moreover, I am pleased to report to the reviewer that the more basic “link test” statistics do indicate good model fit and good specification (_hat P> |z| = .000; _hatsq P> |z| = .571).
In addressing the question of multicollinearity, we used the Stata of command “collin” Which addresses variance inflation tolerance and other such factors that may help a researcher identify potentially problematic variables. For our models the mean variance inflation factor or VIF was 1.11 and there were no variables with a VIF above 1.3.
Finally and addressing yet another good point, the possibility of outliers, we used the visual technique of comparing plotted predicted probabilities against the standardized residuals and found that no particular case was problematic. Of course this process is subject to individual interpretation, but given that there are over 16,000 cases it would be unlikely that any one particular case could create problems.
Thank you again for bringing these elements to our attention, I believe that a presentation of regression diagnostics however brief is an important element of creating and legitimizing any model.
Lastly, I'd like to point out to the reviewer that some of their comments were in line with those comments of the other reviewer. To avoid a degree of redundancy we have addressed those comments in the response to the other reviewers and left this response to be unique. That said, thank you very much for the inclusion of the 2 DOI's that are a valuable addition to the manuscript.
Round 2
Reviewer 2 Report
-I would suggest adding to the literature and referencing it within the introduction and discussion as well. There are studies that have examined intimate partner violence.
1-) https://doi.org/10.1186/s12905-021-01333-1
2-) https://doi.org/10.1371/journal.pone.0275950
It can be published after corrections are made.
Minor editing of English language required
It can be published after corrections are made.